# A CT-Based Clinical, Radiological and Radiomic Machine Learning Model for Predicting Malignancy of Solid Renal Tumors (UroCCR-75)

**DOI:** 10.3390/diagnostics13152548

**Published:** 2023-07-31

**Authors:** Cassandre Garnier, Loïc Ferrer, Jennifer Vargas, Olivier Gallinato, Eva Jambon, Yann Le Bras, Jean-Christophe Bernhard, Thierry Colin, Nicolas Grenier, Clément Marcelin

**Affiliations:** 1Department of Imaging and Interventional Radiology, Hôpital Pellegrin, Place Amélie-Raba-Léon, 33076 Bordeaux, France; 2SOPHiA GENETICS, Multimodal Research, Cité de la Photonique—Bâtiment GIENAH, 11 Avenue de Canteranne, 33600 Pessac, France; lferrer@sophiagenetics.com (L.F.); jvargas@sophiagenetics.com (J.V.); ogallinato@sophiagenetics.com (O.G.); tcolin@sophiagenetics.com (T.C.); 3Department of Urology, Hôpital Pellegrin, Place Amélie-Raba-Léon, 33076 Bordeaux, France

**Keywords:** renal tumors, radiomics, RCC, CT, machine learning

## Abstract

Background: Differentiating benign from malignant renal tumors is important for patient management, and it may be improved by quantitative CT features analysis including radiomic. Purpose: This study aimed to compare performances of machine learning models using bio-clinical, conventional radiologic and 3D-radiomic features for the differentiation of benign and malignant solid renal tumors using pre-operative multiphasic contrast-enhanced CT examinations. Materials and methods: A unicentric retrospective analysis of prospectively acquired data from a national kidney cancer database was conducted between January 2016 and December 2020. Histologic findings were obtained by robotic-assisted partial nephrectomy. Lesion images were semi-automatically segmented, allowing for a 3D-radiomic features extraction in the nephrographic phase. Conventional radiologic parameters such as shape, content and enhancement were combined in the analysis. Biological and clinical features were obtained from the national database. Eight machine learning (ML) models were trained and validated using a ten-fold cross-validation. Predictive performances were evaluated comparing sensitivity, specificity, accuracy and AUC. Results: A total of 122 patients with 132 renal lesions, including 111 renal cell carcinomas (RCCs) (111/132, 84%) and 21 benign tumors (21/132, 16%), were evaluated (58 +/− 14 years, men 74%). Unilaterality (100/111, 90% vs. 13/21, 62%; *p* = 0.02), necrosis (81/111, 73% vs. 8/21, 38%; *p* = 0.02), lower values of tumor/cortex ratio at portal time (0.61 vs. 0.74, *p* = 0.01) and higher variation of tumor/cortex ratio between arterial and portal times (0.22 vs. 0.05, *p* = 0.008) were associated with malignancy. A total of 35 radiomics features were selected, and “intensity mean value” was associated with RCCs in multivariate analysis (OR = 0.99). After ten-fold cross-validation, a C5.0Tree model was retained for its predictive performances, yielding a sensitivity of 95%, specificity of 42%, accuracy of 87% and AUC of 0.74. Conclusion: Our machine learning-based model combining clinical, radiologic and radiomics features from multiphasic contrast-enhanced CT scans may help differentiate benign from malignant solid renal tumors.

## 1. Introduction

Renal cell carcinomas (RCCs) account for approximately 70% of all cases of renal cancer. The most common subtypes are clear-cell renal cell carcinoma (ccRCC), papillary renal cell carcinoma (pRCC) and chromophobe renal cell carcinoma (chRCC), accounting for 70%, 15% and 5% of all RCCs, respectively. As these subtypes have different natural histories and prognoses, it is crucial to differentiate them accurately. Moreover, some common benign tumors show a similar presentation to that of RCCs. Oncocytoma, a benign renal tumor accounting for 5% of all renal masses, is occasionally mistaken for RCC; oncocytomas account for 4–10% of all nephrectomy cases [1]. Furthermore, 5% of angiomyolipomas remain challenging to differentiate from RCC on CT due to their fat-poor nature. This may lead to unnecessary surgical treatment, raising concerns regarding morbidity.

In some circumstances, i.e., when the malignancy status remains unclear before surgery, percutaneous biopsy can be performed. This procedure shows excellent diagnostic performance; it can differentiate between benign and malignant lesions with sensitivity and specificity values of approximately 95%. However, in 20% of cases, the results remain indeterminate, and the complication rate is 8% [2]. Furthermore, for some histological subtypes, pre-operative histological diagnosis is challenging. A recent study reported that 25% of oncocytomas suspected on biopsy were ultimately diagnosed as RCC after surgical removal; 12.5% of these were of the chRCC subtype [3]. There has been an increase in the number of biopsies, particularly on smaller and smaller tumors, with the risk of non-contributory biopsies [4], hence the need to develop imaging characterization.

Multiparametric MRI has been well described in the evaluation of more common subtypes of RCC; however, oncocytoma result in poor imaging diagnostic accuracy [5].

CT is used for renal mass characterization [6]. CT was chosen for the ease of access and spatial resolution for small lesions. In routine clinical practice, qualitative and semi-quantitative parameters are used in combination to distinguish benign from malignant renal masses. Visual analysis of the tumor shape, size, content and enhancement is performed [7]. Some studies showed that an analysis of enhancement patterns on multiphasic contrast-enhanced (MCE)-CT images has high diagnostic accuracy [8]. While enhancement analysis is quantitative, shape and texture analyses remain more subjective and are thus vulnerable to interpretation variability.

Large-scale quantitative parameters can be extracted from medical CT images and then subjected to texture analysis for the detection of local variation in pixel intensity. This has emerged as a novel technique to quantitively evaluate tumor heterogeneity, assess the histopathologic characteristics of carcinomas and help predict prognosis [9,10,11,12].

Radiomics features provide information about the tumor intensity, shape and texture, and application of machine learning analysis to improved imaging data interpretation.

Although recent studies have aimed to differentiate RCCs from benign renal tumors using radiomics [12,13,14], none of them used three-dimensional (3D) radiomic feature extraction combined with clinical and radiological conventional parameters to assess the performance of machine learning (ML) models. The 3D contour-focused segmentation showed a higher stable feature rate [15].

Therefore, the aim of this study was to evaluate the ability of ML models to differentiate between benign and malignant solid renal tumors via the MCE-CT 3D segmentation of extracted radiomic, radiological and clinical features.

## 2. Materials and Methods

### 2.1. Patients

In this retrospective analysis of prospectively collected data, we included all patients who underwent robot-assisted partial nephrectomy for solid renal tumors at our institution between January 2016 and December 2020. Ethics approval was granted by our institutional ethics review board (IRB DR-2013-206). Participants were enrolled from a national kidney cancer database. We included patients who had undergone pre-operative abdominal MCE-CT at our institution. CT examinations performed outside our institution, and those not conducted in accordance with our examination protocol or with low-quality images, were excluded. Patients with missing picture archiving system data were also excluded. Biological, clinical and histological data were extracted from the prospective database. The clinical and biological features analyzed included sex, age and body weight at the time of surgery, the Eastern Cooperative Oncology Group (ECOG) score, the glomerular filtration rate, the presence of urologic symptoms and the clinical tumor–node–metastasis (TNM) stage. Histological findings of interest included malignancy, the histological subtypes of benign and malignant tumors, the Fuhrman grade and the histopathological TNM stage.

### 2.2. CT Examinations

All imaging examinations were performed at our institution using the same 64-slice CT scanner (Optima CT660; GE Healthcare, Milwaukee, WI, USA) before the patients underwent nephrectomy. The CT parameters were as follows: 120 kV; automatic current selection; maximum current, 500 mAs; rotation time, 0.7 s; collimation detector size, 40 × 1.2 mm; field of view, 350 × 350 mm; matrix size, 512 × 512; and reconstruction section thickness, 1.5 mm. For the vascular anatomy analysis, the arterial phase reconstruction pixel size was 0.625 mm.

First, unenhanced CT was performed. Then, a specific enhanced acquisition protocol was applied, including three-phase CT. Nonionic contrast medium (350–400 μmol/L) was injected into the antecubital vein at a rate of 3.5–4 mL/s to a final volume of 80 mL. Arterial phase images were obtained using the scanner’s automatic bolus tracking system (SmartPrep; GE Healthcare), beginning 10 s after the attenuation threshold of 100 UH was reached in the upper abdominal aorta; after an additional 100 s, portal phase images were acquired. Finally, excretory phase images were acquired (10–15 min after injection).

### 2.3. CT

The CT scans were analyzed by one radiologist-in-training (C.G.) with 5 years of image analysis experience. The radiologist was blinded to the clinical and histological findings. The radiological features of all renal tumors were recorded, including infiltration, demarcated contours, homogeneity, calcifications, fat and hemorrhagic components, necrosis, necrotic core, tumor implantation, venous extension, multifocality and bilaterality. Representative cases are shown in Figure 1 and Figure 2.

The enhancement pattern was noted for each lesion. A two-dimensional (2D) region of interest was drawn around the lesion on a single slice for both arterial and portal phase images. A second 2D region of interest was manually drawn on the same slice in a homogenous part of the renal cortex, again for both arterial and portal phase images. Finally, the ratios of the cortical to tumoral intensity values and of the arterial to arterial phases were calculated.

All Digital Imaging and Communication in Medicine images were anonymized. Segmentation was performed by C.G. using SOPHiA DDM for Radiomics v2.1.21 (SOPHiA GENETICS, Saint-Sulpice, Switzerland). In accordance with previous studies, nephrographic phase images were segmented due to their favorable tumor/renal parenchymal contrast. First, the slice on which the tumor was clearest (axial, coronal or sagittal plane) was chosen, and the tumor contours were precisely drawn by hand. Next, a volumetric model of the tumor was constructed using a deformation algorithm. If necessary, the user could manually adjust the semi-automatically obtained contours of the lesion. Each 3D segmentation process took approximately 20 min. The user interface of the segmentation software is shown in Figure 3.

More than 200 radiomic features were automatically extracted from the 3D segmentation model of the tumor (nephrographic phase). Previously described radiomic parameters (shape, pixel intensity and texture features) were analyzed. Dimensionality reduction was then performed using Kendall’s correlation coefficient to avoid redundant parameters. An example process of the radiomic feature extraction is displayed in Figure 4.

To differentiate between benign and malignant solid renal tumors from multimodal (clinical, radiological and radiomic) data with the best predictive performance while ensuring interpretability, the following ML models were trained: logistic regression with LASSO regularization to avoid overfitting (*Logit-LASSO*), binary decision tree (*rpart*), support vector machine with linear kernel (*svmLinear*), bagging method via random forest (*RandomForest*), and boosting method via C5.0 decision tree (*C5.0Tree* and *wC5.0Tree* to deal with imbalanced outcome). Class weights in LASSO-logistic regression and C5.0 tree were also incorporated to deal with the imbalance outcome (*Logit-LASSO* and *wC5.0Tree*, respectively).

Trained models were tested using a 10-fold cross-validation method [16].

The models were compared in terms of their ability to distinguish malignant from benign tumors based on sensitivity, specificity, accuracy and area under the receiver operating characteristic curve (AUC) values.

### 2.4. Statistical Analyses

Clinical data are presented as the mean ± standard deviation for continuous variables and as numbers and percentages for categorial variables. The Bonferroni method was applied for multiple comparisons. Univariable (Wilcoxon and Fisher’s tests) and multivariable (l Logit-LASSO logistic regression, to obtain odds ratios (ORs)) analyses were conducted. All reported *p*-values are two-sided, and *p* < 0.05 was taken to indicate statistical significance.

## 3. Results

### 3.1. Patients and Tumors

A total of 122 patients were included who were surgically treated at our institution between January 2016 and December 2020 (Figure 5). Overall, two had two renal tumors, one had three renal tumors, and one had seven renal tumors; therefore, there were 132 renal lesions in total. There were 111 RCCs: 79 ccRCCs, 16 chRCCs, 13 pRCCs and 3 other rare renal carcinomas. There were also 21 benign lesions: 18 oncocytomas, 2 fat-poor angiomyolipoma (fpAMLs) and 1 other rare benign renal tumor.

The mean age at diagnosis was 58 ± 14 years. Renal tumors were more frequent in males (87/132; 65.9%). Nine patients (7.3%) had sporadic kidney cancer, and seven (5.5%) had a family history of renal cancer. Regarding the bio-clinical characteristics, the mean body mass index was 26.7 kg/m^2^ (range: 17.7–46.9 kg/m^2^) and the mean Cockcroft clearance at diagnosis was 88.4 mL/min. Twenty-five patients (18.9%) had urologic symptoms at diagnosis, and the majority (84.7%) had no physical limitations (ECOG score = 0). Table 1 shows the characteristics of the overall study population as well as of the benign and malignant groups.

Univariable analysis showed that the risk of malignancy was higher in males than females (73.9% vs. 23.8%, *p* = 0.001) and in those with a higher body weight at diagnosis (79.6% vs. 67.4%, *p* = 0.02) (Table 1). In the multivariable analysis, the correlation between sex and malignancy risk remained significant (OR = 2.35). A previous history of cancer (OR = 1.07) also showed a significant correlation with malignancy (Table 2).

### 3.2. Conventional and Enhanced Radiological Features

Malignant lesions were significantly associated with unilaterality (90.1% vs. 61.9%, *p* = 0.02) and necrosis (73% vs. 38.1%, *p* = 0.02) compared with benign lesions. The presence of calcification (30.6% vs. 14.3%, *p* = 0.35), visible fat (9% vs. 19%, *p* = 0.41) and demarcated contours (39.6% vs. 19%, *p* = 0.19) was not significantly different between benign and malignant lesions (Table 3).

In the multivariate analysis, bilaterality (OR = 0.93) was associated with benign lesions, and necrosis showed a trend toward being associated with malignancy (OR = 1.25) (Table 2). Regarding the enhancement pattern, RCCs had a lower tumor/cortex ratio in the portal phase compared with benign tumors (0.61 vs. 0.74, *p* = 0.01), and the difference in the tumor/cortex ratio between the arterial and portal phases was significantly greater for RCCs than benign lesions (−0.22 vs. 0.05, *p* = 0.008). Moreover, the difference in the average lesion intensity between the arterial and portal phases was lower in malignant than benign tumors (0.11 vs. 0.6, *p* = 0.02). (Table 4).

In the multivariable analysis, the difference in the tumor/cortex ratio between the arterial and portal phases was the only enhancement parameter to show a significant difference between RCCs and benign tumors (OR = 0.83) (Table 2).

### 3.3. Three-Dimensional Radiomic Features

More than 200 radiomic parameters were extracted by 3D segmentation in the portal phase. Feature selection based on Kendall’s correlation coefficient led to the retention of 35 radiomic variables in the final analysis. A full list of the retained features is provided in the Appendix A.

In the univariable analyses, five radiomic parameters significantly differentiated malignant from benign tumors: “X90th.discretised.intensity.percentile” (*p* = 0.03), “Global.intensity.peak” (*p* = 0.03), “Intensity.mean.value” (*p* = 0.008), “Local.intensity.peak” (*p* = 0.01) and “volume.at.intensity.fraction.90” (*p* = 0.04) (Table 5). “Intensity.mean.value” was the only radiomic feature that remained significant in the multivariable analysis (OR = 0.99) (Table 2).

### 3.4. Machine Learning Predictive Models

Among the eight ML models, wC5.0Tree showed the best performance according to its sensitivity (95%), specificity (42%), AUC (0.74) and accuracy (87%) (Table 6). Of the 111 malignant lesions, 106 were correctly identified by wC5.0Tree (true positive rate = 0.955), whereas 5 were misclassified as benign (false negative rate = 0.045). Moreover, of the 21 benign lesions, 9 were correctly classified as benign (true negative rate = 0.43), and 12 were misclassified as malignant (false positive rate = 0.57). The most important predictors for the wC5.0Tree algorithm are shown in Figure 6.

## 4. Discussion

This study compared ML models in terms of their ability to differentiate benign from malignant solid renal lesions in patients undergoing pre-operative contrast-enhanced CT at a single institution. The radiomic features were extracted after 3D segmentation of tumors in the nephrographic phase, and eight ML models were trained on all of these features. The C5.0Tree model showed the best predictive performance with a sensitivity of 95%, accuracy of 87% and AUC of 0.74.

Recently, Yap et al. [13] reported a classifier with an AUC of 0.70–0.73 for distinguishing benign from malignant tumors. In that study, radiomic features, shape and texture were analyzed in a large cohort of 735 renal lesions. Thus, combined shape and texture analysis can provide good classification performance. Yang et al. [12] included 118 RCCs and 45 fpAMLs in their study and differentiated them using a radiomic model based on non-contrast CT examinations. An AUC of 0.90 was achieved, while for the analysis of four-phase contrast-enhanced CT images, the AUC was 0.88. Sun et al. [17] compared the ability of ML models to differentiate among several histological subtypes in an analysis of 290 renal tumors. Their classifiers achieved sensitivity values of 86–90% for distinguishing RCCs from benign lesions and ccRCCs from other malignancies; the respective accuracy values were 86% and 90%. Interestingly, the models did not always perform better than trained radiologists. That is the only study in the literature to compare the performance of radiological, radiomic and combined radiological/radiomic models; the model based on enhancement ratios and radiomics performed better than both the radiomic features-only model and expert radiologists. Sun et al. also trained the same support vector machine model to differentiate ccRCCs from pRCCs and chRCCs; ccRCCs from fpAMLs and oncocytomas; and pRCCs and chRCCs from benign lesions.

Erdim et al. [14] achieved very good performance for their random forest algorithm: 84 solid renal masses were correctly identified as malignant at an accuracy of 91.7%. Although this rate is higher than that in our study, they included a smaller patient cohort and artificially adjusted the groups in terms of malignancy to reduce the impact of malignancy on model performance.

The most recent meta-analysis on the use of radiomic features to characterize renal tumors, by Muhlbauer et al. [18], included 30 studies. The overall quality of the studies was relatively low, with a median radiomic quality score of 19.4%. The main reasons for this were insufficient use of feature reduction methods, a lack of internal and external validation and poor data availability. Moreover, ML models using radiomic features are susceptible to overfitting.

Notably, the present study involved all of the common histological subtypes of renal tumors and developed an ML model to differentiate ccRCCs from other malignant subtypes, ccRCCs from fpAMLs and oncocytomas, and all malignant subtypes from benign tumors. In the majority of recent studies, the predictive performance of ML models was tested without distinguishing among subtypes [12,19,20,21,22]. Deng and Yang [12,19] recently differentiated fpAMLs from RCCs (OR = 2.7–4.4). Other studies distinguished oncocytomas from RCCs [23,24,25]. Li et al. developed an ML model to differentiate chRCCs from oncocytomas and achieved very good performance (AUC = 0.964). These models were all developed with the ultimate goal of routine clinical use.

In this study, we used an innovative, semi-automatic 3D segmentation process (in the nephrographic phase) to obtain a volumetric tumor model. Radiomic parameters, including shape, pixel intensity and texture features, were extracted and analyzed. The vast majority of previous radiomic studies used 2D segmentation techniques [12,14,22] and limited data extraction processes, especially those involving heterogeneous tumors. Our semi-automatic method allowed us to analyze tumor margins in one slice and to obtain a volumetric model of the masses via a software-based, time-efficient deep learning algorithm.

Our study had several limitations. First, the cohort was imbalanced; malignant and benign tumors represented 84% and 16% of all tumors, respectively. We included patients who underwent surgical treatment for the removal of a renal lesion; before partial nephrectomy, the images were reviewed, and most of the benign lesions were excluded. Furthermore, fpAMLs accounted for only 1.5% of all solid renal tumors in our cohort, which is lower than the rates (6–13%) reported in other recent studies [13,17,19]. Furthermore, we only included patients who underwent pre-operative CT examination at our institution using a protocol specific for MCE-CT. This allowed us to obtain homogenous imaging data, thus increasing the robustness of the analysis of radiomic features. However, this also raises issues concerning the clinical relevance and generalizability of the results. Furthermore, the 3D segmentation technique automatically reduces the margins of the volumetric model by 1 mm. This method is widely used in radiomic studies for margin shrinkage, which frequently described manual reductions in tumor contours of 1–3 mm [14,21,22,26], because it reduces the vulnerability of radiomic features to partial volume effects. However, Kocak et al. [27] recently showed that segmentation had a non-negligible impact on radiomic features. Their “specific contour” segmentation method yielded AUC values of 0.85–0.98 when distinguishing benign from malignant tumors compared with 0.75–0.8 when using the margin shrinkage approach. Since this is time consuming, it is not adapted to current clinical practice at the moment.

Finally, our findings were not validated against external data. Regarding the relatively small number of renal tumors included in the cohort, we trained and evaluated the models using data from the entire cohort and the 10-fold cross-validation technique. Although this method is widely used in radiomic studies, larger cohorts and external validation sets should be used to assess the performance of the ML algorithm developed herein.

## 5. Conclusions

This study showed that ML models can help with the non-invasive differentiation of malignant from benign solid renal tumors. These classifiers can efficiently analyze clinical, conventional radiological and radiomic features extracted from MCE-CT images to help clinicians diagnose and treat renal tumors.

## Figures and Tables

**Figure 1 diagnostics-13-02548-f001:**
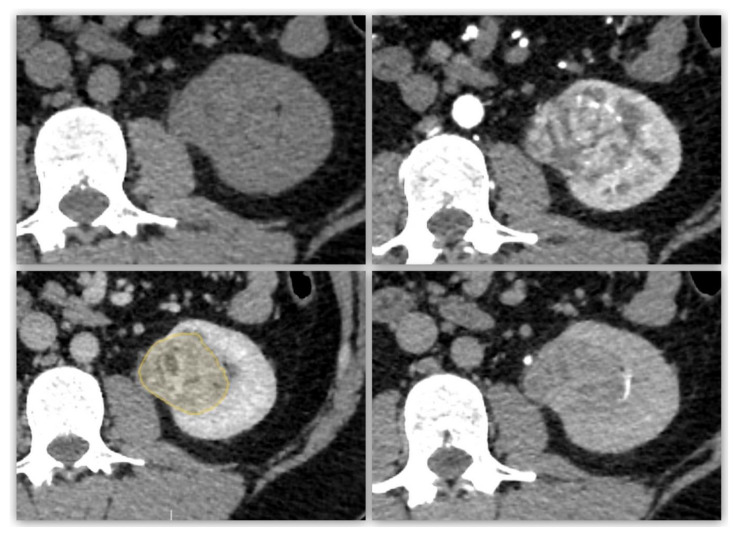
Representative case of segmentation for a papillary renal cell carcinoma (pRCC). A 68-year-old female with a pRCC. Unenhanced CT images (**upper left**), arterial phase (**upper right**), nephrographic time (**lower left**) and excretory phase (**lower right**) show a well-defined, demarcated, with heterogenous enhancement. Yellow outlined drawing represents segmentation margins.

**Figure 2 diagnostics-13-02548-f002:**
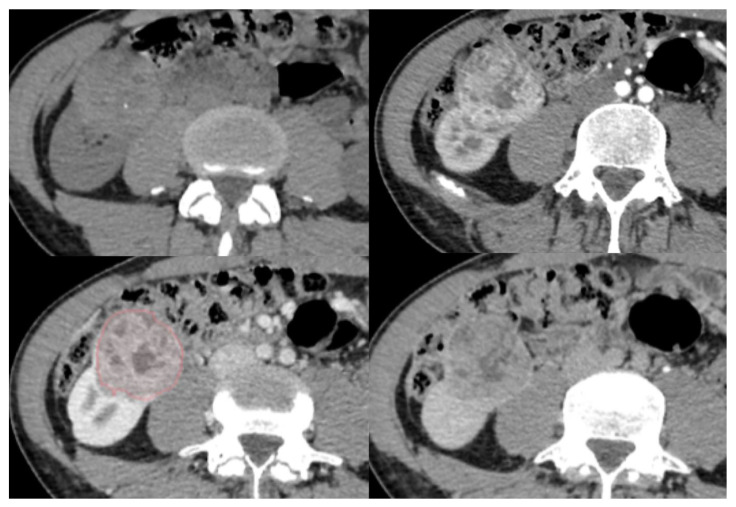
Representative case of segmentation for a clear cell renal cell carcinoma (ccRCC). A 55-year-old man with a demarcated, well-defined, homogenous tumor, with a regular calcification, diagnosed as a ccRCC, at multiphasic contrast-enhanced CT scan: unenhanced (**upper left**), arterial (**upper right**), nephrographic (**lower left**) and excretory (**lower right**) phases. Red outline represents tumor segmentation.

**Figure 3 diagnostics-13-02548-f003:**
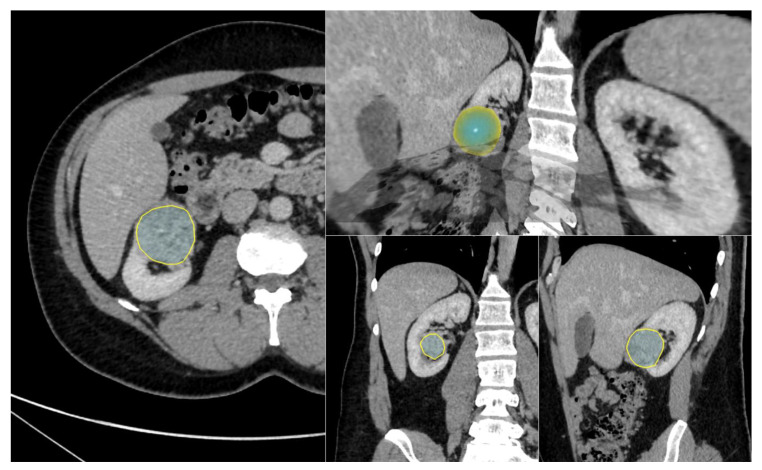
Three-dimensional (3D) segmentation software interface for a clear cell RCC. Tumor segmentation on 2D slice at nephrographic phase (yellow outline, **left**), with corresponding 2D segmentations on coronal and sagittal plans (**lower right**), and a volumic model of the 3D segmentation (**upper right**) used for radiomic features extraction.

**Figure 4 diagnostics-13-02548-f004:**
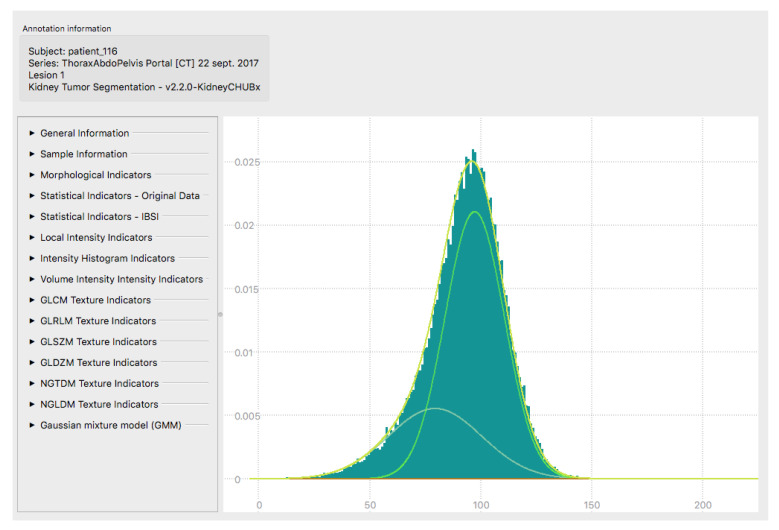
Representative histogram for radiomic features extraction using software interface. IBSI = Image Biomarker Standardization Initiative; GLCM = Gray-Level Co-occurrence Matrix; GLRLM = Gray-Level Run Length Matrix; GLSZM = Gray-Level Size Zone Matrix; GLDZM = Gray-Level Distance Zone Matrix; NGTDM = Neighborhood Gray Tone Difference Matrix; NGLDM = Neighborhood Gray-Level Dependence Matrix.

**Figure 5 diagnostics-13-02548-f005:**
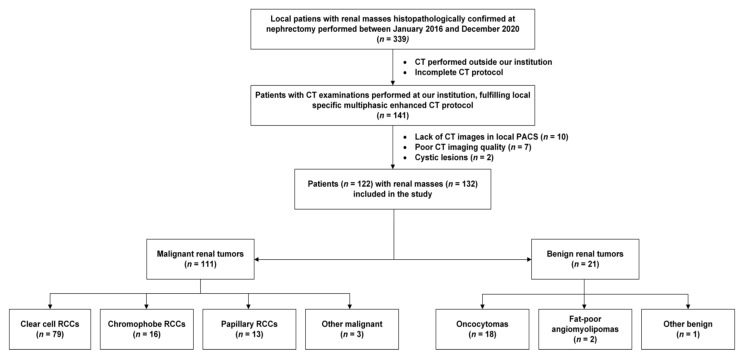
Flow-chart of the study. CT = computed tomography; PACS = Picture Archiving and Communication System; RCC = renal cell carcinoma. Numbers of patients and renal lesions are in parentheses.

**Figure 6 diagnostics-13-02548-f006:**
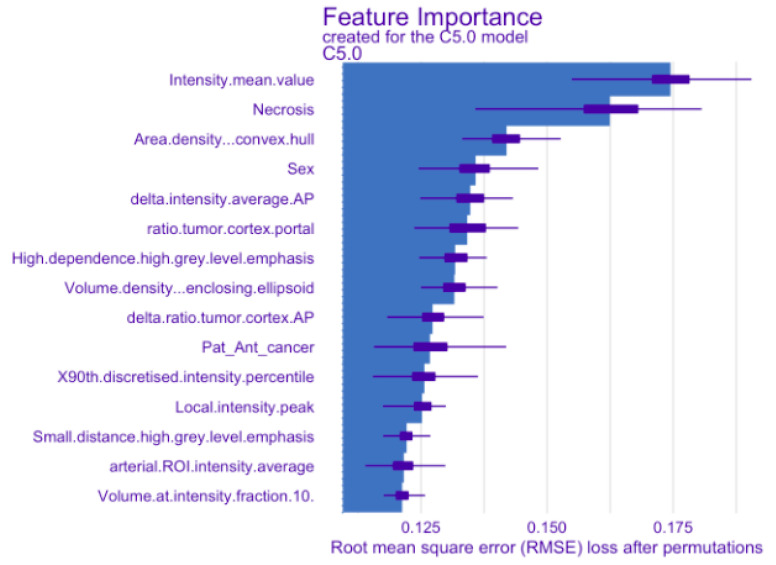
Plot graph of variables involved in prediction model. The 15 most important features in tumor classification were plotted on the graph in order of their importance. The variable that played the most important role in prediction was “intensity mean value”, which corresponds to the average value of the intensities observed in the volumic segmentation at the portal time.

**Table 1 diagnostics-13-02548-t001:** Characteristics of the study population.

Variable	Benign (n = 21)	Malignant (n = 111)	Total (n = 132)	Adjusted *p*-Value
Sex				**0.001**
Female	16 (76.2%)	29 (26.1%)	45 (34.1%)	
Male	5 (23.8%)	82 (73.9%)	87 (65.9%)	
Cockcroft clearance				0.061
Missing data	0	1	1	
Mean (SD)	71.229 (40.212)	91.758 (39.730)	88.467 (40.366)	
Age				0.099
Missing data	1	1	2	
Mean (SD)	64.450 (11.834)	57.327 (14.273)	58.423 (14.122)	
Body weight				**0.025**
Mean (SD)	67.476 (16.366)	79.604 (18.391)	77.674 (18.567)	
BMI				0.119
Mean (SD)	24.843 (4.968)	27.041 (5.567)	26.691 (5.518)	
ECOG score				0.351
Missing data	0	8	8	
0	20 (95.2%)	85 (82.5%)	105 (84.7%)	
1–3	1 (4.8%)	18 (17.5%)	19 (15.3%)	
Cancer history				0.099
No	20 (95.2%)	81 (73.0%)	101 (76.5%)	
Yes	1 (4.8%)	30 (27.0%)	31 (23.5%)	
Symptoms at diagnosis				0.240
No	20 (95.2%)	87 (78.4%)	107 (81.1%)	
Yes	1 (4.8%)	24 (21.6%)	25 (18.9%)	
Pathological stage *				0.103
T1	11 (52.4%)	84 (75.7%)	95 (72.0%)	
T2/T3	10 (47.6%)	27 (24.3%)	37 (28.0%)	
Family history of renal cancer				0.371
Missing data	0	5	5	
Yes	0	7 (6.6%)	7 (5.5%)	
No	21 (100%)	99 (93.4%)	120 (94.5%)	

BMI, body mass index; ECOG, Eastern Cooperative Oncology Group. * Based on the American Joint Committee on Cancer TNM staging system, 8th edition. Adjusted *p*-values < 0.05 are in bold.

**Table 2 diagnostics-13-02548-t002:** Multivariable analysis of factors differentiating benign and malignant solid renal tumors. The coefficients are given for the covariates selected by the Logit-LASSO procedure.

Variable Name	Coefficient	Odds Ratio (IC)	*p*
(Intercept)	−0.999	0.368	
delta.ratio.tumor.cortex.AP	−0.192	0.826 (0.057, 1)	0.682
Bilateral lesion—Yes (ref: No)	−0.071	0.931 (0.583, 1)	0.332
Necrosis—Yes (ref: No)	0.223	1.250 (1, 6.355)	0.646
Intensity.mean.value	−0.006	0.994 (0.983, 1)	0.640
Sex—Men (ref: Women)	0.853	2.346 (1.183, 4.384)	0.987
History_cancer—Yes (ref: No)	0.069	1.072 (1, 29.045)	0.650

The 95% confidence intervals were estimated using percentile approach, and the *p*-values were obtained as the rate of non-selection of the feature, both over 1000 bootstrap replicates.

**Table 3 diagnostics-13-02548-t003:** Conventional radiological features.

Variable Name	Benign (n = 21)	Malignant (n = 111)	Total (n = 132)	Adjusted *p*-Value
Bilateral.lesion				**0.024**
No	13 (61.9%)	100 (90.1%)	113 (85.6%)	
Yes	8 (38.1%)	11 (9.9%)	19 (14.4%)	
Calcification				0.348
No	18 (85.7%)	77 (69.4%)	95 (72.0%)	
Yes	3 (14.3%)	34 (30.6%)	37 (28.0%)	
Contour.regularity				0.185
Demarcated	17 (81.0%)	67 (60.4%)	84 (63.6%)	
Infiltrating	4 (19.0%)	44 (39.6%)	48 (36.4%)	
Fat				0.409
No	17 (81.0%)	101 (91.0%)	118 (89.4%)	
Yes	4 (19.0%)	10 (9.0%)	14 (10.6%)	
Homogeneous				0.714
No	14 (66.7%)	80 (72.1%)	94 (71.2%)	
Yes	7 (33.3%)	31 (27.9%)	38 (28.8%)	
Monofocal.lesion				0.099
No	8 (38.1%)	17 (15.3%)	25 (18.9%)	
Yes	13 (61.9%)	94 (84.7%)	107 (81.1%)	
Necrosis				**0.025**
No	13 (61.9%)	30 (27.0%)	43 (32.6%)	
Yes	8 (38.1%)	81 (73.0%)	89 (67.4%)	
Necrotic.core				0.099
No	16 (76.2%)	56 (50.5%)	72 (54.5%)	
Yes	5 (23.8%)	55 (49.5%)	60 (45.5%)	

Adjusted *p*-values < 0.05 are in bold.

**Table 4 diagnostics-13-02548-t004:** Enhancement features.

Variable Name	Benign (n = 21)	Malignant (n = 111)	Total (n = 132)	Adjusted *p*-Value
arterial.ROI.intensity.average	0	3	3	0.680
Mean (SD)	105.033 (47.859)	97.828 (42.567)	99.001 (43.355)	
ratio.tumor.cortex.arterial	0	3	3	0.562
Missing data	0	3	3	
Mean (SD)	0.809 (0.361)	0.847 (0.299)	0.841 (0.309)	
ratio.tumor.cortex.portal	0	1	1	**0.014**
Missing data	0	1	1	
Mean (SD)	0.740 (0.157)	0.612 (0.177)	0.632 (0.180)	
delta.intensity.average.AP	0	3	3	**0.024**
Missing data	0	3	3	
Mean (SD)	0.600 (1.275)	0.107 (0.537)	0.187 (0.727)	
delta.ratio.tumor.cortex.AP	0	3	3	**0.008**
Missing data	0	3	3	
Mean (SD)	0.055 (0.492)	−0.217 (0.309)	−0.172 (0.357)	

Adjusted *p*-values < 0.05 are in bold.

**Table 5 diagnostics-13-02548-t005:** Three-dimensional radiomic features.

Variable Name	Benign (n = 21)	Malignant (n = 111)	Total (n = 132)	Adjusted *p*-Value
X90th.discretized.intensity.percentile	22.906 (3.446)	20.435 (4.551)	20.828 (4.475)	**0.025**
Area.density…aligned.bounding.box	0.544 (0.054)	0.532 (0.042)	0.534 (0.044)	0.680
Area.density…convex.hull	1.045 (0.086)	1.027 (0.053)	1.030 (0.060)	0.933
Area.density…oriented.bounding.box	0.572 (0.047)	0.561 (0.036)	0.563 (0.038)	0.680
Center.of.mass.shift.cm.	0.076 (0.061)	0.112 (0.113)	0.106 (0.107)	0.360
Cluster.shade	−91.862 (274.983)	−80.688 (251.410)	−82.466 (254.236)	0.680
Correlation	0.675 (0.134)	0.655 (0.149)	0.658 (0.146)	0.714
Flatness	0.776 (0.105)	0.763 (0.104)	0.765 (0.104)	0.680
Global.intensity.peak	163.910 (40.505)	140.206 (38.455)	143.977 (39.597)	**0.025**
Gray.level.variance..GLDZM.	28.405 (8.614)	24.046 (7.029)	24.739 (7.442)	0.105
High.dependence.high.gray.level.emphasis	14,606.667 (10,457.428)	11,350.721 (5853.078)	11,868.712 (6847.744)	0.632
High.dependence.low.gray.level.emphasis	0.167 (0.295)	0.479 (1.360)	0.429 (1.257)	0.140
Intensity.histogram.coefficient.of.variation	0.214 (0.084)	0.210 (0.064)	0.210 (0.067)	1.000
Intensity.mean.value	118.633 (43.093)	83.399 (25.630)	89.005 (31.661)	**0.008**
Intensity.based.interquartile.range..Original.Data.	43.410 (14.082)	41.619 (15.683)	41.904 (15.402)	0.599
Inverse.elongation	0.860 (0.070)	0.861 (0.096)	0.861 (0.092)	0.680
Large.distance.low.gray.level.emphasis	0.204 (0.215)	0.428 (0.665)	0.392 (0.621)	0.099
Local.intensity.peak	135.500 (48.979)	100.684 (40.422)	106.223 (43.608)	**0.016**
Max.value	227.190 (50.281)	242.550 (169.494)	240.106 (156.655)	0.180
Min.value..Original.Data.	−54.714 (59.865)	−68.604 (44.202)	−66.394 (47.051)	0.714
Number.of.compartments.GMM.	3.333 (1.390)	3.333 (1.231)	3.333 (1.252)	1.000
Number.of.gray.levels	218.524 (75.883)	219.757 (102.488)	219.561 (98.485)	0.714
Skewness..Original.Data.	−0.258 (0.587)	0.022 (1.025)	−0.023 (0.972)	0.105
Small.distance.emphasis	0.496 (0.135)	0.433 (0.115)	0.443 (0.120)	0.099
Small.distance.high.gray.level.emphasis	167.005 (50.245)	135.899 (62.734)	140.847 (61.810)	0.051
Small.distance.low.gray.level.emphasis	0.005 (0.005)	0.004 (0.004)	0.004 (0.004)	0.714
Small.zone.emphasis	0.586 (0.033)	0.570 (0.031)	0.573 (0.032)	0.099
Spherical.disproportion	1.128 (0.105)	1.113 (0.112)	1.115 (0.111)	0.680
Volume.at.intensity.fraction.10.	0.998 (0.004)	0.999 (0.003)	0.999 (0.003)	0.714
Volume.at.intensity.fraction.90.	0.003 (0.006)	0.001 (0.002)	0.002 (0.003)	0.042
Volume.density…aligned.bounding.box	0.466 (0.035)	0.462 (0.042)	0.463 (0.041)	0.919
Volume.density…enclosing.ellipsoid	0.976 (0.014)	0.975 (0.022)	0.975 (0.021)	0.680
Volume.density…oriented.bounding.box	0.504 (0.027)	0.501 (0.036)	0.502 (0.035)	0.919
Volume.fraction.difference.between.intensity.fractions	0.995 (0.009)	0.997 (0.004)	0.997 (0.005)	0.105
Zone.size.entropy	6.610 (0.303)	6.598 (0.337)	6.600 (0.330)	0.919

Adjusted *p*-values < 0.05 are in bold. Data are means (standard deviation). GLDZM, gray-level distance zone matrix; GMM, Gaussian mixture model.

**Table 6 diagnostics-13-02548-t006:** Predictive performance of the machine learning-based models.

Model	Accuracy	Sensitivity	Specificity	Precision	Brier Score	F1 Score	AUC
Rpart	0.895	0.983	0.429	0.901	0.098	0.94	0.608
C5.0Tree	0.861	0.956	0.362	0.888	0.117	0.921	0.736
Logit-Lasso	0.855	0.966	0.267	0.874	0.119	0.918	0.721
RandomForest	0.879	0.972	0.386	0.893	0.105	0.931	0.773
svmLinear	0.852	0.954	0.314	0.88	0.106	0.916	0.81
wRpart	0.765	0.815	0.5	0.896	0.19	0.854	0.654
wC5.0Tree	0.867	0.95	0.424	0.897	0.114	0.923	0.739
wllasso	0.811	0.865	0.524	0.906	0.159	0.885	0.705

SVM, support vector machine; AUC, area under the receiver operating characteristic curve.

## Data Availability

Data sharing is not applicable to this article.

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
