# Peer review of "A CT-Based Clinical, Radiological and Radiomic Machine Learning Model for Predicting Malignancy of Solid Renal Tumors (UroCCR-75)"

_diagnostics, 2023, doi:10.3390/diagnostics13152548_

Round 1

Reviewer 1 Report

dear author

thank you for this article which is perfectly in the scope of this special issue dedicated to quantitative imaging.

It is a complex article, with a lot of data and statistical analysis. I do not see a statistician in the authors or in the acknowledgements. Who did the stats? 

The main limitations of this study are the low number of benign tumours and the somewhat disappointing results on the multi-area analyses. We can see that there is still a long way to go to differentiate between benign and malignant tumours

These are my remarks:

- General: the ORs provided as they are, are not comprehensible, please give each time the 95% CI and the p

- abstract: to be reviewed at a distance once the corrections of the article are made. Please note that making the difference between benign and malignant remains complicated even with radiomics and machine learning.

- keyword: missing

- 2.1: the ethics section, please add the IRB ethics number and delete "requirement for written consent was waived".

- 2.1, l 105: "low image quality": what does it mean?

- 2.3, l 130: put "one radiologist (5 years of experiment)" instead

- figure 2: the lower left image is not in the same plane as the others (coronal instead of axial), can you put an axial image?

- figure 3: you call figure 5 but I think this is a mistake and that you should call figure 2

- 2.6: you used "10 fold cross validation". why? can you give a reference? 

- Result. 3.1: review the beginning with the flowchart because it is not clear. I will just put that 122 patients were included in the end and refer to the flowchart for details.

- table 2: add IC 95 and p

- 3.2: the concept of enhancement is not clear. does it mean that malignant tumours have a higher wash out than benign ones? can you explain a little bit because it is confusing and we don't understand this difference which seems to be important.

- figure 7, l 263, the t in "the" is missing

- l274: it is difficult to see the significance of an OR at 0.99, add CI and p.

3.4: what were the selection criteria? why this one rather than another with a better AUC or accuracy? we do not understand the choice. Is there a possibility to compare them with a p?

- limitations: add that it is very time consuming and not adapted to current clinical practice at the moment

- conclusion: It is important to emphasize that making the difference between benign and malignant remains complicated even with radiomics and machine learning.

Thank you for your interest in my remarks. I look forward to reading your corrected version

good work.

Author Response

thank you for this article which is perfectly in the scope of this special issue dedicated to quantitative imaging.

It is a complex article, with a lot of data and statistical analysis. I do not see a statistician in the authors or in the acknowledgements. Who did the stats? 

Authors: Statistics were performed by statistician Jennifer Vargas

The main limitations of this study are the low number of benign tumours and the somewhat disappointing results on the multi-area analyses. We can see that there is still a long way to go to differentiate between benign and malignant tumours

 Authors:thanks for the comment. Benign tumors are less frequent.

These are my remarks:

- General: the ORs provided as they are, are not comprehensible, please give each time the 95% CI and the p

Variable name

Coefficient

Odds ratio (IC)

p

(Intercept)

-0.999

0.368

delta.ratio.tumor.cortex.AP

-0.192

0.826 (0.057, 1)

0.682

Bilateral lesion - Yes (ref : No)

-0.071

0.931 (0.583, 1)

0.332

Necrosis - Yes (ref : No)

0.223

1.250 (1, 6.355)

0.646

Intensity.mean.value

-0.006

0.994 (0.983, 1)

0.640 

Sex - Men (ref : Women)

0.853

2.346 (1.183, 4.384)

0.987

History_cancer - Yes (ref : No)

0.069

1.072 (1, 29.045)

0.650

Authors: thanks for the comment. We added the 95%CI and the p value:

- abstract: to be reviewed at a distance once the corrections of the article are made. Please note that making the difference between benign and malignant remains complicated even with radiomics and machine learning.

Authors: Thanks for the comment, we modify the conclusion accordingly

“may help differentiating benign from malignant solid renal tumors”

- keyword: missing

Authors: Sorry , we add the keywords:

renal tumors; radiomics; RCC; CT; machine learning

- 2.1: the ethics section, please add the IRB ethics number and delete "requirement for written consent was waived".

Authors: Thanks for the comment.

We modify accordingly

(IRB  DR-2013-206).

- 2.1, l 105: "low image quality": what does it mean?

Authors: Low image quality mean artifact due to patient movement or contrast acquisition.

- 2.3, l 130: put "one radiologist (5 years of experiment)" instead

Authors: thanks for the comment, we modified it accordingly.

- figure 2: the lower left image is not in the same plane as the others (coronal instead of axial), can you put an axial image?

Authors:Thanks for the comment, we change for an axial plane image.

- figure 3: you call figure 5 but I think this is a mistake and that you should call figure 2

Authors: thanks for the comment, we modified it accordingly.

- 2.6: you used "10 fold cross validation". why? can you give a reference? 

Authors: This is the standard method of cross validation on literature.

10.1186/s13244-021-01115-1

- Result. 3.1: review the beginning with the flowchart because it is not clear. I will just put that 122 patients were included in the end and refer to the flowchart for details.

Authors: Thanks, we modified it accordingly.

122 patients included finally between January 2016 and December 2020 surgically treated at our institution (Figure 5).

- table 2: add IC 95 and p

Authors: See answer below.

- 3.2: the concept of enhancement is not clear. does it mean that malignant tumours have a higher wash out than benign ones? can you explain a little bit because it is confusing and we don't understand this difference which seems to be important.

Authors: Thanks for the comment. We modify for a better understanding.

Regarding the enhancement pattern, RCCs had a lower tumor/cortex ratio in the portal phase compared with benign tumors (0.61 vs. 0.74, p = 0.01) and the difference in the tumor/cortex ratio between the arterial and portal phases was significantly greater for RCCs than benign lesions (-0.22 vs. 0.05, p = 0.008). Moreover, the difference in the average lesion intensity between the arterial and portal phases (washout) was lower in malignant than benign tumors (0.11 vs. 0.6, p = 0.02). (Table 4).

In the multivariate analysis, the difference in the tumor/cortex ratio between the arterial and portal phases was the only enhancement parameter to show a significant difference between RCCs and benign tumors (OR = 0.83)

- figure 7, l 263, the t in "the" is missing

Authors: Sorry but the is well written.

- l274: it is difficult to see the significance of an OR at 0.99, add CI and p.

Authors: à see the answer below à 95% CI [0.983, 1] à replace the sentence “Intensity.mean.value” was the only radiomic feature that remained significant in the multivariate analysis (OR = 0.99) by “Intensity.mean.value” was the only radiomic feature that was selected in the multivariable analysis (OR = 0.99, 95% CI [0.983, 1])

3.4: what were the selection criteria? why this one rather than another with a better AUC or accuracy? we do not understand the choice. Is there a possibility to compare them with a p?

Authors à The wC5.0Tree model was retained because 1) it globally achieved good performance (e.g., top 3 AUC with random forest and support vector machine) and the interpretability analysis (feature importance and PDPs) showed that the model did not overfit the data, which was unclear for the other tested models.

- limitations: add that it is very time consuming and not adapted to current clinical practice at the moment

Authors: Thanks, we modified it accordingly.

- conclusion: It is important to emphasize that making the difference between benign and malignant remains complicated even with radiomics and machine learning.

 Authors: Thanks, we modified it accordingly.

This study showed that ML models can help for non-invasive differentiation of malignant from benign solid renal tumors

Thank you for your interest in my remarks. I look forward to reading your corrected version

good work.

Reviewer 2 Report

The paper "A CT-based clinical, radiological and radiomic machine learning model for predicting malignancy of solid renal tumors" requires some revisions to improve its clarity and technical focus. While the authors provide some clinical context, the paper would benefit from a more thorough explanation of the machine learning approaches used and the technical results obtained.

Firstly, the authors should provide a more detailed explanation of the machine learning classification approaches used in the study. While the paper mentions the use of machine learning algorithms, it does not provide a clear justification for why these specific approaches were chosen or how they were implemented. The authors should provide a brief overview of the algorithms and explain how they were used in the study to help readers better understand the technical side of the study.

Secondly, the authors should provide more detailed explanations of the results obtained, including any limitations or areas for further improvement.

Thirdly, the figures in the paper should be improved for clarity and informativeness. The figures provided in the paper are blurry and difficult to interpret, making it difficult for readers to understand the results. The authors should ensure that the figures are clear and accurate, and that they provide sufficient information to support the results presented in the paper.

Finally, the authors should consider the target audience when revising the paper. If the paper is intended for a technical audience, the authors should focus more on the technical aspects of the study, including detailed explanations of the machine learning approaches and results. If the paper is intended for a medical audience, the authors should provide more clinical context and explain the implications of the study for patient care.

In summary, the paper would benefit from revisions that focus on improving its technical focus and clarity, including more detailed explanations of the machine learning approaches and results, improved figures, and consideration of the target audience.

Author Response

The paper "A CT-based clinical, radiological and radiomic machine learning model for predicting malignancy of solid renal tumors" requires some revisions to improve its clarity and technical focus. While the authors provide some clinical context, the paper would benefit from a more thorough explanation of the machine learning approaches used and the technical results obtained.

Firstly, the authors should provide a more detailed explanation of the machine learning classification approaches used in the study. While the paper mentions the use of machine learning algorithms, it does not provide a clear justification for why these specific approaches were chosen or how they were implemented. The authors should provide a brief overview of the algorithms and explain how they were used in the study to help readers better understand the technical side of the study.

Authors: Thanks for this comment, we added this sentence:

To differentiate between benign and malignant solid renal tumors from multimodal (clinical, radiological and radiomic) data with the best predictive performance while ensuring interpretability, the following ML models were trained: logistic regression with LASSO regularization to avoid overfitting (Logit-LASSO), binary decision tree (rpart), support vector machine with linear kernel (svmLinear), bagging method via random forest (RandomForest), and boosting method via C5.0 decision tree (C5.0Tree and wC5.0Tree to deal with imbalanced outcome). Class weights in LASSO-logistic regression and C5.0 tree were also incorporated to deal with imbalance outcome (Logit-LASSO and wC5.0Tree, respectively).

Secondly, the authors should provide more detailed explanations of the results obtained, including any limitations or areas for further improvement.

Authors: Results are detailed on text and tables. Limitations are written at the end of discussion.

Thirdly, the figures in the paper should be improved for clarity and informativeness. The figures provided in the paper are blurry and difficult to interpret, making it difficult for readers to understand the results. The authors should ensure that the figures are clear and accurate, and that they provide sufficient information to support the results presented in the paper.

Authors: Thanks for this comment. We deleted figures 6 and 7.

Finally, the authors should consider the target audience when revising the paper. If the paper is intended for a technical audience, the authors should focus more on the technical aspects of the study, including detailed explanations of the machine learning approaches and results. If the paper is intended for a medical audience, the authors should provide more clinical context and explain the implications of the study for patient care.

Authors: Thanks for this comment. We added clinical context in the introduction:

An increase in the number of biopsies, particularly on smaller and smaller tumors, with the risk of non-contributory biopsies (4), hence the need to develop imaging characterization.

Multiparametric MRI has been well described in the evaluation of more common subtypes of RCC, however oncocytoma resulting in poor imaging diagnostic accuracy

In summary, the paper would benefit from revisions that focus on improving its technical focus and clarity, including more detailed explanations of the machine learning approaches and results, improved figures, and consideration of the target audience.

Authors: Thanks for this comment, we added this sentence:

To differentiate between benign and malignant solid renal tumors from multimodal (clinical, radiological and radiomic) data with the best predictive performance while ensuring interpretability, the following ML models were trained: logistic regression with LASSO regularization to avoid overfitting (Logit-LASSO), binary decision tree (rpart), support vector machine with linear kernel (svmLinear), bagging method via random forest (RandomForest), and boosting method via C5.0 decision tree (C5.0Tree and wC5.0Tree to deal with imbalanced outcome). Class weights in LASSO-logistic regression and C5.0 tree were also incorporated to deal with imbalance outcome (Logit-LASSO and wC5.0Tree, respectively).

We change figure 2.

Reviewer 3 Report

1. Missing keywords.

2. Lack of innovation, with little involvement in the machine learning solutions mentioned in the article, and only a simple accumulation of terminology.

3. The article layout is too careless, and some chapter titles may have extra positions.

4. Only introduced comparative experiments related to diagnosis, rarely involving the innovation of the proposed solution.

5. The content is all comparative experiments, and the logic is basically carried out according to one's own comparative experiments. The entire length of the article introduces one's own comparative experiments, without a clear explanation of one's writing innovation.

1.缺少关键字。

2.缺乏创新,很少涉及文章中提到的机器学习解决方案,只有简单的术语积累。

3.文章布局太粗心了,有些章节标题可能有多余的位置。

4.只介绍了相关诊断的对比实验,很少涉及创新性的解决方案的提出。

5.内容都是比较实验,逻辑基本上都是按照自己的比较实验来进行的。文章通篇介绍自己的对比实验,没有明确说明自己的写作创新。

Author Response

  1. Missing keywords.

Authors: Sorry, we add the keywords:

renal tumors; radiomics; RCC; CT; machine learning

  1. Lack of innovation, with little involvement in the machine learning solutions mentioned in the article, and only a simple accumulation of terminology.

Authors: Thanks for this comment, we modified the introduction

“None of them used three-dimensional (3D) radiomic feature extraction combined with clinical and radiological conventional parameters to assess the performance of machine learning (ML) models. 3D contour-focused segmentation showed a higher stable feature rate”

  1. The article layout is too careless, and some chapter titles may have extra positions.

Authors: Thanks for this comment, we deleted chapters on M&M

  1. Only introduced comparative experiments related to diagnosis, rarely involving the innovation of the proposed solution.

Authors: Thanks for this comment, we added this sentence:

3D contour-focused segmentation showed a higher stable feature rate (14).

  1. The content is all comparative experiments, and the logic is basically carried out according to one's own comparative experiments. The entire length of the article introduces one's own comparative experiments, without a clear explanation of one's writing innovation.
    Authors: Thanks for this comment, we added this sentence:

An increase in the number of biopsies, particularly on smaller and smaller tumors, with the risk of non-contributory biopsies (4), hence the need to develop imaging characterization.

Round 2

Reviewer 3 Report

Agree to receive